# FEATURE-FREE APPROACH FOR SAT SOLVER SELECTION

## ABSTRACT

Boolean Satisfiability Problem is a cornerstone in computer science and artificial intelligence, underpinning numerous applications through its ability to solve complex computational problems. However, existing SAT solvers face significant limitations, including the complexity and domain expertise required for feature design, the static nature of many feature sets that limit adaptability to evolving problem structures, and the poor generalization of handcrafted features to new instances, thereby constraining performance across diverse SAT problem distributions. To address these challenges, we introduce a Feature-Free SAT Solver Selection (F2S3), which integrates the Sensitive-Associative Cascade Forest (SACF), Correlation Refinement Factor Graph (CRFG), and Dual-Proximity Graph Representation (DPGR) to address the complexities of SAT problems. F2S3 method transforms problem instances into graph data, utilizes CRFG to maintain the higher-order nature of the graph structure and node relationships, and uses DPGR to enhance the graph data features and map them to low-dimensional vectors. This approach effectively captures the structural intricacies of graph data and improves feature representation in low-dimensional spaces, overcoming the limitations of previous methods regarding feature sparsity and generalization ability. Experiments conducted on datasets from the ASlib database demonstrate that F2S3 outperforms existing solutions, particularly in scenarios where previous methods were hindered by challenges such as feature sparsity and computational inefficiency. The method's performance is evaluated across multiple competitive datasets, showing high gap values and consistent robustness.

## 1 INTRODUCTION

Given a Boolean formula, the Boolean satisfiability (SAT) problem determines whether there exists a satisfying assignment. As the first problem recognized as NP-complete, the SAT problem exerts significant influence in the fields of computer science and artificial intelligence. Numerous problems from various domains—including logic, graph theory, operations research, automated planning, formal verification, and more—can be transformed into SAT problems or require the use of a SAT oracle Alyahya et al. (2022). Consequently, any advancements in SAT solving can create a ripple effect across these related fields. Despite its intrinsic complexity, the development of the Davis–Putnam–Logemann–Loveland (DPLL) algorithm Davis & Putnam (1960); Govindasamy et al. (2024), along with the subsequent Conflict-Driven Clause Learning (CDCL) framework Zhang & Malik (2002); Marques-Silva et al. (2021), has enabled current state-of-the-art solvers to effectively handle SAT instances with millions of variables.

While SAT solving is a hallmark of symbolic artificial intelligence, recent years have witnessed a revolutionary impact on computer science from another branch of artificial intelligence: data-dependent algorithms or machine learning approaches. This raises a natural question: Can machine learning enhance SAT solving? There are two main lines of research that can be explored to address this problem. The relatively new line, which has attracted more attention recently, is the end-to-end SAT solving, exemplified by NeuroSAT Selsam et al. (2019). NeuroSAT is an experimental SAT solver that learns to solve SAT problems after being trained as a classifier to predict satisfiability. It utilizes a message-passing neural network to predict the satisfiability of random SAT problems and learns to search for satisfying assignments to explain that bit of supervision. When it guesses satisfiable, the satisfying assignment can often be decoded from its activations. However, the scale

and complexity of the problems addressed by this method are relatively small, and it often exhibits unstable performance on instances that deviate from the distribution of its training data.

The second line of research, leveraging machine learning to improve CDCL SAT solving, is more practical in applications. Key approaches include SAT algorithm configuration and algorithm selection. The former involves automatically identifying parameter settings that optimize a given SAT solver's performance on a specific set or distribution of problem instances, while the latter refers to selecting the most suitable solver from a "portfolio" of different algorithms for a given SAT instance. This paper focuses on the latter. Although creating a single, universally effective SAT solver seems intuitive, it is well known in the community that no one solver dominates all others across diverse problem instances. SAT Solver Selection (SSS) Kerschke et al. (2019); Alissa et al. (2023) exploits machine learning techniques to dynamically select the most appropriate solver and thereby combine the complementary strengths of different algorithms across problem instances to improve overall performance. By dynamically selecting the most appropriate solver based on instance-specific features, SSS can substantially enhance solving efficiency. SATzilla Xu et al. (2008) is the first successful and best known portfolio-based SAT solver. Its initial version innovatively employed ridge regression techniques to accurately predict the efficiency of the solver when dealing with unknown SAT instances. Leveraging this technological edge, SATzilla has won multiple gold medals in SAT competitions, setting a significant milestone in the development of the SAT solving field. Additionally, methods such as 3S Kadioglu et al. (2011), MapleCOMSPS Liang et al. (2016), and Kissat_MAB Cherif et al. (2021) have also emerged as winners in relevant competitions. Lingeling ayv algorithmBiere (2014) outperformed 34 other solvers on 300 benchmark instances in the 2014 SAT competition with a 77% completion rate. GraSSZhang et al. (2024) builds upon traditional SAT solver selection methods by incorporating graph structures, enabling more effective use of the structural information in SAT problems with promising results. In addition to SSS, algorithm selection has also been successfully applied in areas like the DelfiKatz et al. (2018), QBF PortfolioHoos et al. (2018), and automated machine learning (AutoML)Feurer et al. (2018).

A critical limitation in the existing SSS methods is their heavy reliance on handcrafted features, which are manually designed based on expert knowledge. While these features can be effective, they introduce several challenges. First, the process of designing features is often complex and requires domain expertise. Second, many feature sets have remained unchanged for years, limiting their adaptability to evolving problem structures. This limited adaptability is sometimes compounded by the non-negligible cost of computing such features, especially when dealing with large or complex instances. Third, handcrafted features may not generalize well to novel instances, thereby constraining model performance across diverse SAT problem distributions. To address these challenges, feature-free approaches that automatically learn hidden structures from raw problem instances have emerged as a promising alternative. By directly capturing complex patterns without manual feature engineering, feature-free methods reduce the reliance on expert knowledge and improve model generalization across varied SAT instances.

This paper proposes a Feature-Free method for SAT Solver Selection, F2S3, which integrates Correlation Refinement Factor Graph, Dual-Proximity Graph Representation, and Sensitive-Associative Cascade Forest. By leveraging deep graph embedding techniques and an advanced cascade forest model, our method optimizes both the representation of SAT instances and the decision-making process in solver selection. This results in a more flexible and effective algorithm selection strategy that is capable of automatically adapting to problem variations and providing more accurate SAT-solving recommendations.

The major contributions of this work are as follows:

- We propose F2S3, a feature-free SAT solver selection method that eliminates manual feature engineering, mitigates graph sparsity, and improves prediction accuracy and robustness.

- We introduce Correlation Refinement Factor Graph, Dual-Proximity Graph Representation, and the Sensitivity-Correlation Cascaded Forest to enhance graph representation and decision accuracy, addressing sparsity and refining ensemble decision-making.

- Experimental results show that F2S3 outperforms existing methods, improving selection accuracy and adaptability across various task settings.

## 2 RELATED WORK

### 2.1 GRAPH NEURAL NETWORKS IN SAT SOLVING

Graph neural networks (GNNs) have recently attracted attention in the SAT community, as Boolean formulas can be naturally represented as graphs. NeuroSAT Selsam et al. (2019) first demonstrated the potential of message-passing networks to capture satisfiability patterns directly from formula structure. Later work extended this idea with NsNet Li & Si (2022), which incorporated probabilistic inference, and GraSS Zhang et al. (2024), which introduced heterogeneous representations with task-sensitive objectives. These studies show that graph-based models can complement traditional handcrafted statistics. They highlight the promise of GNNs in learning structural properties of SAT instances, while also pointing to opportunities for closer integration with efficient CDCL-based solving procedures.

### 2.2 SAT SOLVER SELECTION

Solver selection has become an important topic in SAT research. SATzilla Xu et al. (2008) pioneered performance prediction using instance-level features, which inspired a variety of subsequent approaches, including cost-sensitive clustering Malitsky et al. (2013), nearest-neighbor strategies Nikolić et al. (2013), and automated machine learning pipelines Malone et al. (2017). More recent frameworks such as Sunny-as2 Liu et al. (2021) refine feature selection, solver scheduling, and pre-solver strategies, achieving strong results across benchmarks. While these approaches have demonstrated substantial success, they typically rely on handcrafted features. It led to excessive computation time and hindered further improvement in generalization due to the lack of introducing new features, which motivates exploration of representation learning methods to remove the dependence on manual feature engineering in this study.

### 2.3 GRAPH EMBEDDING AND CASCADE FOREST

The main idea of this study is to integrate graph embedding and deep random forest to achieve feature-free SAT solver selection. Graph embedding provides a framework for learning low-dimensional representations of graph-structured data, using techniques such as random walks, matrix factorization, and deep autoencoders Zheng et al. (2023). These methods capture structural relationships, which are especially beneficial for SAT, where formulas are represented as bipartite or heterogeneous graphs. The Node-Similarity Factor Graph Liu et al. (2023) is particularly effective for SAT due to its ability to preserve both local and global structural dependencies. Random Forest Rigatti (2017) leverages an ensemble of decision trees for prediction, while Cascade Forest Zhou & Feng (2019) enhances prediction accuracy through hierarchical optimization.

## 3 APPROACH

This section introduces our approach, and Figure 1 illustrates the overall workflow.

### 3.1 PROBLEM DEFINITION

The satisfiability (SAT) problem involves deciding whether a satisfying assignment exists for a Conjunctive Normal Form (CNF) formula as $F = c_1 \wedge c_2 \wedge \cdots \wedge c_m$, where each clause $c_i$ is a disjunction of literals, and each literal is either a Boolean variable $x$ or its negation $\neg x$. Given a fixed set of SAT solvers $\{A_1, A_2, \ldots, A_K\}$, the problem of *SAT Solver Selection* is to construct a mapping from a given SAT problem instance to its optimal solver. The task of this paper is to use feature-free method to learn this mapping by leveraging a training set of SAT instances paired with their optimal solvers, enabling the model to predict the most suitable solver for unseen instances during the test phase.

### 3.2 CORRELATION REFINEMENT FACTOR GRAPH CONSTRUCTION

Inspired by the Node-Similarity Factor Graph Liu et al. (2023), we construct an initial factor graph for each SAT instance based on its Conjunctive Normal Form (CNF) representation. The resulting

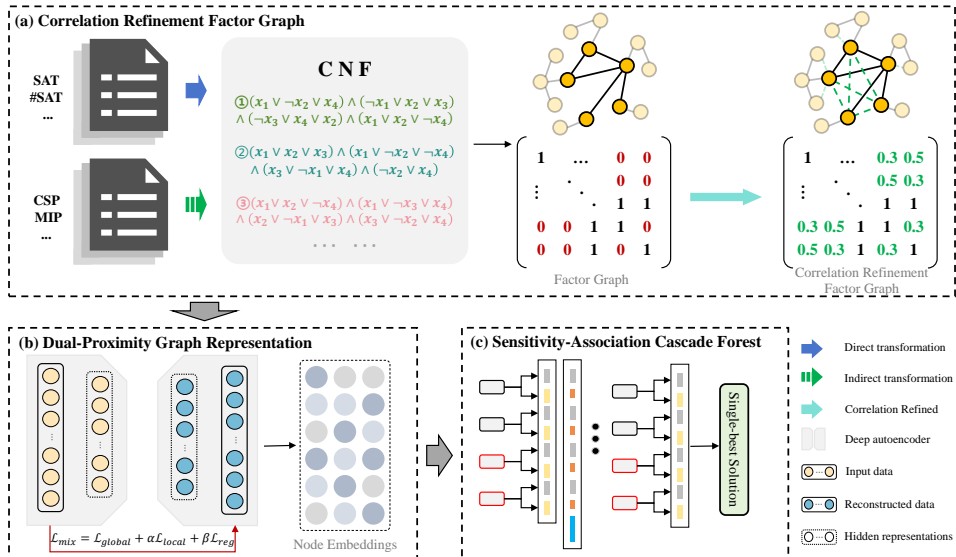

Figure 1: Overview of the proposed F2S3 framework. A SAT instance is first transformed into a Correlation Refinement Factor Graph (CRFG), then mapped into low-dimensional embeddings via the Dual-Proximity Graph Representation (DPGR), and finally fed into the Sensitive-Associative Cascade Forest (SACF) to predict the optimal solver.

factor graph consists of two types of nodes: variable nodes $x_j$ corresponding to the Boolean variables, and factor nodes $c_i$ corresponding to the clauses. An edge is established between a variable node and a factor node whenever the variable $x_j$ or its negation $\neg x_j$ appears in the clause $c_i$. This graph is represented by an adjacency matrix $\mathbf{S} \in \{0,1\}^{m \times n}$, where the entry $S_{ij}$ is 1 if the variable $x_j$ or its negation $\neg x_j$ appears in clause $c_i$, and 0 otherwise.

However, the adjacency matrix $\mathbf{S}$ is often highly sparse, as each clause $c_i$ involves only a small subset of variables. Consequently, many variable and clause nodes lack direct connections and are only related through long indirect paths. This confines information propagation to local neighborhoods, weakens the capture of global dependencies, and ultimately leads to insufficient graph representations. Therefore, we introduce the *Correlation Refinement Factor Graph* (CRFG), which refines the original factor graph to alleviate sparsity while maintaining its structural integrity, as detailed in the following.

**Factor node correlation optimization.** For each factor node $c_i$, we define its associated set of variable nodes as $L(c_i) = \{x_j \mid x_j \in c_i \text{ or } \neg x_j \in c_i\}$. If two factor nodes $c_i$ and $c_k$ share at least one variable, i.e., $L(c_i) \cap L(c_k) \neq \varnothing$, we introduce a *direct similarity* defined by

$$P_{c_i,c_k}^{\text{direct}} = \frac{|L(c_i) \cap L(c_k)|}{\sqrt{|L(c_i)| \cdot |L(c_k)|}}.$$

Moreover, if two factor nodes are not directly connected but are indirectly related through other nodes, we define an *indirect similarity* as

$$P_{c_i,c_k}^{\text{indirect}} = \frac{N(c_i) \cdot N(c_k)}{\|N(c_i)\| \cdot \|N(c_k)\|},$$

where $N(c_i)$ and $N(c_k)$ denote the neighborhood vectors of factor nodes $c_i$ and $c_k$, respectively.

**Variable node correlation extension.** For each variable node $x_j$, we define its associated factor set as $C(x_j) = \{c_i \mid x_j \in c_i \text{ or } \neg x_j \in c_i\}$. If two variables $x_p$ and $x_q$ co-occur in common clauses, i.e., $C(x_p) \cap C(x_q) \neq \varnothing$, they are connected with a *direct similarity*:

$$P_{x_p,x_q}^{\text{direct}} = \frac{|C(x_p) \cap C(x_q)|}{\sqrt{|C(x_p)| \cdot |C(x_q)|}}.$$

For variables without direct co-occurrence but with similar neighborhoods, we introduce an *indirect similarity* defined as

$$P^{\text{indirect}}_{x_p, x_q} = \frac{N(x_p) \cdot N(x_q)}{\|N(x_p)\| \cdot \|N(x_q)\|},$$

where $N(x_p)$ and $N(x_q)$ denote the neighborhood vectors of variables $x_p$ and $x_q$, respectively.

**Adjacency matrix optimization.** Finally, we integrate the direct and indirect similarities of both factor nodes and variable nodes to obtain the optimized bipartite adjacency matrix:

$$\mathbf{S}' = \lambda \mathbf{P}_{\text{direct}} + \mu \mathbf{P}_{\text{indirect}},$$

where $\lambda$ and $\mu$ are trade-off parameters balancing local and global similarity. Compared to the original matrix $\mathbf{S} \in \{0, 1\}^{m \times n}$, the optimized matrix $\mathbf{S}' \in \mathbb{R}^{m \times n}$ is denser, significantly reduces zero elements, and enhances latent correlations among nodes.

### 3.3 DUAL-PROXIMITY GRAPH REPRESENTATION

After obtaining the optimized factor graph from the *Correlation Refinement Factor Graph (CRFG)*, we derive a refined bipartite adjacency matrix $\mathbf{S}' \in \mathbb{R}^{m \times n}$ that encodes richer structural correlations between clauses and variables. While $\mathbf{S}'$ alleviates the sparsity inherent in the original factor graph, it still represents a purely structural enhancement and does not directly yield compact feature representations suitable for downstream learning tasks. To bridge this gap, we propose the *Dual-Proximity Graph Representation (DPGR)*. DPGR takes $\mathbf{S}'$ as input, employs a deep auto-encoder to learn low-dimensional embeddings, and jointly integrates global and local proximity constraints. This ensures that the learned representations preserve local pairwise correlations while simultaneously capturing global structural dependencies.

**Auto-encoder representation learning.** Given the refined bipartite adjacency matrix $\mathbf{S}' \in \mathbb{R}^{m \times n}$, each row $\mathbf{z}_i^{(c)} \in \mathbb{R}^n$ corresponds to the neighborhood vector of a clause $c_i$, and each column $\mathbf{z}_j^{(x)} \in \mathbb{R}^m$ corresponds to the neighborhood vector of a variable $x_j$. For notational simplicity, we use $\mathbf{z}_i$ to denote the adjacency vector of a node, which can be either a clause or a variable. These vectors serve as the input to a deep auto-encoder that maps them into a low-dimensional latent space.

The encoder applies a sequence of nonlinear transformations:

$$\mathbf{y}_i^{(1)} = \sigma\big(W^{(1)}\mathbf{z}_i + b^{(1)}\big), \quad \mathbf{y}_i^{(k)} = \sigma\big(W^{(k)}\mathbf{y}_i^{(k-1)} + b^{(k)}\big), \quad k = 2, \ldots, K,$$

where $\mathbf{y}_i^{(k)}$ denotes the hidden representation of node $i$ at the $k$-th layer, and $\sigma(\cdot)$ is a nonlinear activation function. $W^{(k)}$ and $b^{(k)}$ are the weight matrix and bias vector of the $k$-th encoder layer, respectively. The decoder reconstructs the original adjacency vector, producing $\hat{\mathbf{z}}_i$, which is compared with $\mathbf{z}_i$ to compute the reconstruction error.

**Local proximity preservation.** To preserve local structural correlations, we introduce a local proximity loss. If two nodes exhibit strong correlation in the optimized adjacency matrix $\mathbf{S}'$, their embeddings should be closely aligned in the latent space. Formally:

$$\mathcal{L}_{local} = \sum_{i,j} S'_{ij} \|\mathbf{y}_i^{(K)} - \mathbf{y}_j^{(K)}\|_2^2,$$

where $S'_{ij}$ denotes the correlation strength between clause $c_i$ and variable $x_j$ derived from CRFG.

**Global proximity preservation.** To capture global structural dependencies, DPGR incorporates a global proximity loss based on the reconstruction error of the auto-encoder. Even if two nodes are not directly connected, their embeddings should be close if they share similar neighborhood structures:

$$\mathcal{L}_{global} = \sum_{i=1}^{m} \|\hat{\mathbf{z}}_i - \mathbf{z}_i\|_2^2,$$

where $\mathbf{z}_i$ is the input adjacency vector and $\hat{\mathbf{z}}_i$ is its reconstruction. This constraint ensures that global topological patterns are preserved in the latent space.

**Regularization.** To improve generalization and stabilize training, we add Frobenius norm penalties on both encoder and decoder weights. Let $W^{(k)}$ and $b^{(k)}$ denote the parameters of the $k$-th encoder layer, and $\widehat{W}^{(k)}$ and $\widehat{b}^{(k)}$ those of the corresponding decoder layer. The regularization term is defined as

$$\mathcal{L}_{reg} = \frac{1}{2} \sum_{k=1}^{K} \left( \|W^{(k)}\|_F^2 + \|\widehat{W}^{(k)}\|_F^2 \right),$$

where $\|\cdot\|_F$ denotes the Frobenius norm. This term constrains parameter magnitudes, reduces overfitting, and encourages smoother latent representations.

**Overall objective.** The complete optimization objective integrates direct proximity, indirect proximity, and regularization as follows:

$$\mathcal{L} = \mathcal{L}_{global} + \alpha \cdot \mathcal{L}_{local} + \beta \cdot \mathcal{L}_{reg},$$

where $\alpha$ and $\beta$ are hyperparameters controlling the balance among different loss components. The optimization procedure of DPGR follows the paradigm of Structural Deep Network Embedding Wang et al. (2016), and the detailed derivations are provided in the Appendix A.1.1.

After training, the decoder generates the reconstructed adjacency vector $\hat{\mathbf{z}}_i$ for each node. These reconstructed adjacency vectors represent the learned graph structure in the latent space and are used for evaluating the model's reconstruction quality. The learned node embeddings from the encoder, which summarize the structural properties of the entire SAT instance, are then used as the input to the subsequent supervised learning module.

### 3.4 SENSITIVE-ASSOCIATIVE CASCADE FOREST

We propose the *Sensitive-Associative Cascade Forest* (SACF), a supervised learning module built on the cascade forest Zhou & Feng (2019). SACF incorporates *feature sensitivity optimization* and *feature correlation enhancement* to mitigate the bias of conventional splitting criteria and explicitly capture feature interactions. As illustrated in Figure 2, the instance embedding $\mathbf{z}$ is progressively refined into class distributions, sensitivity-adjusted distributions, correlation-enhanced vectors, and updated inputs, until producing the final solver prediction $\hat{y}$.

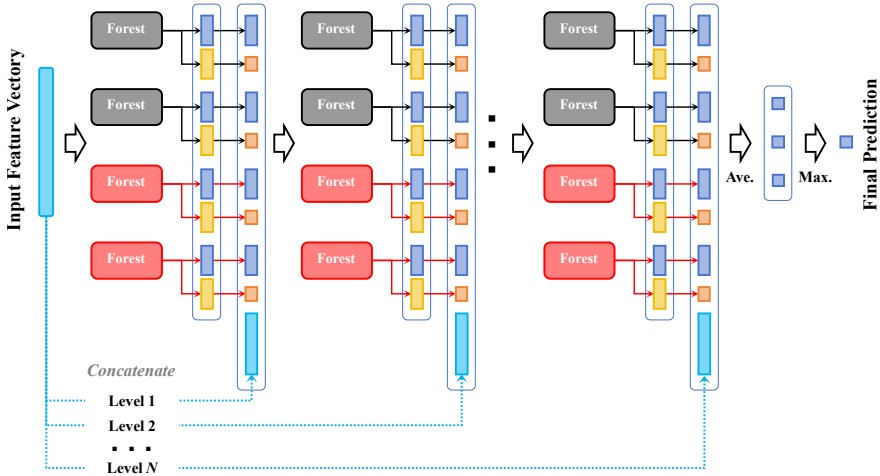

Figure 2: Architecture of the Sensitive-Associative Cascade Forest (SACF). Each layer includes two random forests (black) and two completely random forests (red). Outputs are optimized by feature sensitivity (yellow) and correlation enhancement (brown).

**Layer-wise class distribution.** At the $\ell$-th level, each forest $\mathcal{F}_m^{(\ell)}$ produces a class distribution vector:

$$\mathbf{p}^{(\ell,m)}(\mathbf{z}) = (p_1^{(\ell,m)}, \dots, p_K^{(\ell,m)}),$$

where $p_k^{(\ell,m)}$ denotes the probability that solver $A_k$ is optimal for the input $\mathbf{z}$. Concatenating the outputs of all forests gives:

$$\mathbf{v}^{(\ell)}(\mathbf{z}) = \text{concat}_m \, \mathbf{p}^{(\ell,m)}(\mathbf{z}).$$

**Feature sensitivity optimization.** The Gini index in decision tree splitting tends to favor high-cardinality but less informative features and fails to explicitly capture interactions among dimensions, which weakens the utilization of discriminative signals. To address this issue, SACF introduces a feature sensitivity optimization mechanism, where the term "feature" refers to embedding dimensions derived from the learned representation rather than handcrafted SAT statistics. This mechanism identifies critical dimensions by measuring their frequency in splitting chains and explicitly reinforces their contribution in the leaf-node class distributions.

Specifically, each decision tree generates a *splitting chain*, i.e., the sequence of dimensions used from the root to the leaf. Based on all splitting chains, the sensitivity score of dimension $f_j$ is defined as

$$s_{f_j} = \frac{1}{|\mathcal{T}|} \sum_{t \in \mathcal{T}} \mathbf{1}[f_j \in \text{chain}(t)],$$

where $\mathcal{T}$ denotes the set of trees in the forest and $\mathbf{1}[\cdot]$ is the indicator function. The resulting vector $\mathbf{s} = (s_{f_1}, \ldots, s_{f_d})$ quantifies the relative importance of each embedding dimension. The process is illustrated in Figure 3, where the original cascade forest paths (red) are complemented with additional feature-sensitive branches (blue).

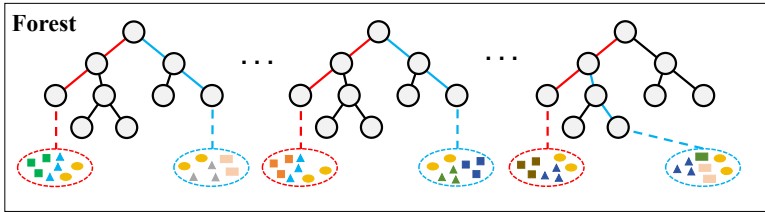

Figure 3: Feature sensitivity optimization in SACF. Red paths represent the original cascade forest, and blue paths denote the optimization branches.

For an input instance $\mathbf{z}$, suppose its corresponding class distribution at a leaf node is $\mathbf{p}$. SACF generates alternative paths dominated by highly sensitive dimensions and obtains a set of auxiliary class distributions $\{\hat{\mathbf{p}}^{(1)}, \hat{\mathbf{p}}^{(2)}, \ldots\}$. The optimized distribution is then obtained by averaging the original and auxiliary distributions:

$$\tilde{\mathbf{p}} = \text{mean}\left(\{\mathbf{p}\} \cup \{\hat{\mathbf{p}}^{(1)}, \hat{\mathbf{p}}^{(2)}, \ldots\}\right).$$

**Feature correlation enhancement.** If two features $f_i, f_j$ co-occur in the same splitting chain, they are considered correlated. The correlation strength is defined as:

$$C_{ij} = \frac{\#\{\text{chains containing } f_i, f_j\}}{\#\{\text{chains}\}},$$

which yields a correlation matrix $\mathbf{C} \in \mathbb{R}^{d \times d}$.

We first select the top-$k$ most sensitive features $\{f_1, \ldots, f_k\}$ according to $\mathbf{s}$, and then retrieve their most correlated partners $\{f'_1, \ldots, f'_k\}$ from $\mathbf{C}$. For each pair, we compute the mean squared deviation (MSD) of their class distributions, forming the correlation-enhanced vector:

$$\mathbf{q}^{(\ell)}(\mathbf{z}) = \frac{1}{k} \sum_{i=1}^{k} \left(\mathbf{p}_{f_i} - \mathbf{p}_{f'_i}\right)^2.$$

This operation not only captures feature interactions but also reduces sensitivity to the parameter $k$.

**Input update and cascade training.** The input to the next level is the concatenation of the previous representation, the optimized class distribution, and the correlation-enhanced vector:

$$\mathbf{h}^{(\ell)}(\mathbf{z}) = [\,\mathbf{h}^{(\ell-1)}(\mathbf{z}), \mathbf{v}^{(\ell)}(\mathbf{z}), \mathbf{q}^{(\ell)}(\mathbf{z})\,], \quad \mathbf{h}^{(0)}(\mathbf{z}) = \mathbf{z}.$$

Each layer is trained using cross-validation estimates to prevent overfitting, and the cascade continues to grow until validation performance no longer improves.

**Final prediction.** During inference, the final prediction is made by aggregating the outputs of all forests at the last layer $L$:

$$\hat{y} = \arg\max_k \frac{1}{M} \sum_{m=1}^{M} p_k^{(L,m)},$$

where $M$ denotes the number of forests in the last level. The predicted label $\hat{y}$ corresponds to the optimal solver from the pool $\{A_1, \ldots, A_K\}$.

## 4 EXPERIMENTAL RESULTS

We train and evaluate our approach on seven ASlib scenarios Bischl et al. (2016), selected for their overlap with the Open Algorithm Selection Challenge (OASC) Lindauer et al. (2017) and sunny-as2 Liu et al. (2021), covering SAT (*Sora*, *Svea*), MaxSAT (*Magnus*, *Monty*), CSP (*Caren*, *Camilla*), and MIP (*Mira*). Performance is measured using the *gap* metric relative to the virtual best solver. Details are in Appendix A.2.

### 4.1 MAIN RESULTS

To assess the effectiveness of the F2S3 model, we evaluate it against several baselines on seven ASlib scenarios. As shown in Table 1, the baselines include AS-ASL and AS-RF Malone et al. (2017), ASAP.v3 Gonard et al. (2017), star-zilla Xu et al. (2012), sunny-based variants like sunny-as2-fk Liu et al. (2021) and sunny-autok Lindauer et al. (2019), and the neural model NeuroSAT (modified to a multi-class classifier for SAT solver selection) Selsam et al. (2019).

Table 1: Gap values for different scenarios of comparative experiments.

| Baselines | Sora | Svea | Magnus | Monty | Caren | Camilla | Mira | Avg. |
|---|---|---|---|---|---|---|---|---|
| AS-ASL | −0.6692 | 0.4385 | −1.0528 | −6.3895 | −1.7325 | 0.4385 | −0.4065 | −1.3391 |
| AS-RF | −0.3700 | 0.5853 | −1.0521 | −6.8992 | −1.0617 | 0.5853 | 0.4947 | −1.1025 |
| ASAP.v3 | 0.0639 | _0.6881_ | 0.4963 | _0.7631_ | 0.3276 | _0.6881_ | _0.5091_ | _0.5052_ |
| star-zilla | 0.1706 | 0.1715 | _0.5751_ | 0.1731 | −0.6409 | 0.1715 | 0.0328 | 0.0934 |
| Sunny-autok | 0.0021 | 0.5789 | 0.4924 | 0.6318 | _0.6440_ | 0.5789 | −0.0137 | 0.4163 |
| sunny-as2-fk | **0.3428** | 0.6643 | 0.4458 | 0.5846 | 0.0845 | N/A | −0.1891 | 0.4244 |
| NeuroSAT | 0.2831 | 0.5922 | 0.3379 | 0.4620 | 0.3782 | 0.4216 | 0.1928 | 0.3811 |
| **F2S3 (ours)** | _0.3374_ | **0.7786** | **0.6283** | **0.9230** | **0.8373** | **0.8921** | **0.6280** | **0.7178** |

*Note: Bold indicates the best performance, and underline indicates the second best in each column.*

Table 1 summarizes the comparative results on seven ASlib scenarios. Our proposed **F2S3** achieves the best overall performance, with an average gap value of $0.7178$, substantially outperforming the strongest baseline ASAP.v3 ($0.5052$). On most individual scenarios, F2S3 also attains the highest gap values, including *Magnus* ($0.6283$), *Monty* ($0.9230$), *Svea* ($0.7786$), *Caren* ($0.8373$), *Camilla* ($0.8921$), and *Mira* ($0.6280$). Notably, the gains in *Monty*, *Svea*, and *Mira* are particularly large compared with all baselines, underscoring the robustness of F2S3 across diverse datasets.

The superior performance of F2S3 can be interpreted from three perspectives. First, compared with traditional solver selection methods (e.g., AS-ASL, AS-RF, Sunny-autok), which rely on manually designed SAT features, F2S3 is entirely feature-free. By constructing refined factor graphs and learning solver-oriented embeddings automatically, our approach avoids costly feature engineering and achieves more consistent performance across different problem distributions. Second,

relative to learning-based baselines such as NeuroSAT, F2S3 consistently yields higher gap values (e.g., 0.9230 vs. 0.4620 on *Monty*, 0.8373 vs. 0.3782 on *Caren*). This highlights the advantage of our CRFG+DPGR pipeline for extracting structure-preserving representations, together with SACF for capturing discriminative patterns that generic graph neural architectures overlook. Finally, although the experiments are conducted on SAT and MaxSAT scenarios, the feature-free design of F2S3 makes it broadly applicable. Preliminary studies on CSP and MIP datasets show similarly strong improvements, indicating that F2S3 generalizes beyond Boolean satisfiability to a wide range of combinatorial optimization domains. Together, these results demonstrate that F2S3 not only achieves state-of-the-art solver selection accuracy but also provides a general and extensible framework.

## 4.2 Ablation Study

We conduct ablation studies on seven ASlib scenarios to evaluate the contribution of each component in F2S3. All results are reported as average precision over ten independent runs, with detailed per-run results provided in Appendix A.3.1. As shown in Table 2, the full model consistently achieves the best performance across all scenarios. Removing SACF leads to precision drops of 3–6 points, with the largest degradation observed on *Mira* ($-6.54$), highlighting its role in enhancing discriminative capability through feature sensitivity and correlation modeling. Removing DPGR causes even larger decreases, up to $-8.66$ on *Mira*, demonstrating its importance in preserving global-local structural information and learning robust embeddings.

The most severe performance deterioration occurs when both DPGR and SACF are removed, with precision drops exceeding 10 points in several scenarios (e.g., *Sora* and *Monty*). These results indicate that the two modules are not only individually effective but also complementary: DPGR preserves structural information while SACF refines embeddings for solver prediction. Overall, the ablation results clearly confirm that both modules are essential for achieving the high precision of F2S3.

Table 2: Ablation results of F2S3 on seven ASlib scenarios.

| Variants | Sora(%) | Svea(%) | Magnus(%) | Monty(%) | Caren(%) | Camilla(%) | Mira(%) |
|---|---|---|---|---|---|---|---|
| **F2S3** | **50.18** | **81.66** | **80.62** | **93.72** | **83.04** | **88.52** | **78.58** |
| *w/o SACF* | 44.22 ↓5.96 | 76.82 ↓4.84 | 75.32 ↓5.30 | 89.84 ↓3.88 | 79.46 ↓3.58 | 83.28 ↓5.24 | 72.04 ↓6.54 |
| *w/o DPGR* | 43.88 ↓6.30 | 76.82 ↓4.84 | 74.56 ↓6.06 | 89.12 ↓4.60 | 79.32 ↓3.72 | 81.12 ↓7.40 | 69.92 ↓8.66 |
| *w/o DPGR+SACF* | 39.32 ↓10.86 | 71.66 ↓10.00 | 72.00 ↓8.62 | 82.64 ↓11.08 | 75.52 ↓7.52 | 78.36 ↓10.16 | 69.06 ↓9.52 |

## 4.3 Discussion

This study presents a feature-free approach for SAT solver selection, addressing the limitations of GraSS (which depends on unreleased feature computation methods) and NeuroSAT (which focuses primarily on local node-edge interactions). Our approach eliminates manual feature engineering, automatically captures structural information, and mitigates performance degradation due to feature sparsity or distribution shifts. As a result, it achieves improved generalization and stability across various scenarios. The key components-CRFG, DPGR and SACF-complement each other in alleviating graph sparsity, learning robust embeddings, and enhancing discriminative power. These combined strengths enable F2S3 to outperform existing methods, even when individual components alone are less effective. Additionally, the parameter sensitivity analysis in Appendix A.3.2 shows that F2S3 is robust to variations in embedding dimensions and hyperparameters, demonstrating its practical applicability without extensive parameter tuning.

## 5 Conclusion

We propose F2S3, a feature-free approach for SAT solver selection that integrates Correlation Refinement Factor Graph, Dual-Proximity Graph Representation, and Sensitive-Associative Cascade Forest. F2S3 outperforms existing methods, particularly in scenarios impacted by feature sparsity and computational inefficiency, demonstrating its effectiveness and robustness across diverse problem instances.

## ETHICS STATEMENT

We adhere to the ICLR Code of Ethics. No human subjects were involved in our research, and all datasets used are publicly available. We have ensured that our research complies with ethical guidelines, including privacy, fairness, and avoiding harmful applications. There are no conflicts of interest or commercial sponsorship.

## REPRODUCIBILITY STATEMENT

The model used in this research is available at https://anonymous.4open.science/r/F2S3-5228/, and the datasets are from the public dataset ASlib.

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

## THE USE OF LARGE LANGUAGE MODELS (LLMS)

We did not use any Large Language Models (LLMs) for research ideation or writing in the development of this paper. All research ideas, analysis, and writing were carried out by the authors without the assistance of LLMs.

## A  APPENDIX

### A.1  APPROACH DETAILS

#### A.1.1  DPGR OPTIMIZATION.

The optimization of *Dual-Proximity Graph Representation* (DPGR) aims to jointly preserve both local and global structural properties while preventing overfitting. The overall loss is defined as

$$\mathcal{L} = \mathcal{L}_{global} + \alpha \cdot \mathcal{L}_{local} + \beta \cdot \mathcal{L}_{reg},$$

where $\mathcal{L}_{global}$ enforces global proximity preservation, $\mathcal{L}_{local}$ maintains local pairwise correlations, and $\mathcal{L}_{reg}$ controls model complexity through parameter regularization. The parameters of the encoder and decoder are optimized via backpropagation.

**Gradient computation.**  Let $\theta = \{W^{(k)}, \widehat{W}^{(k)}\}_{k=1}^{K}$ denote the learnable parameters of the auto-encoder. The optimization objective is to minimize $\mathcal{L}$ with respect to $\theta$. The gradients of the loss with respect to the decoder and encoder parameters are:

$$\frac{\partial \mathcal{L}}{\partial \widehat{W}^{(k)}} = \frac{\partial \mathcal{L}_{global}}{\partial \widehat{W}^{(k)}} + \beta \frac{\partial \mathcal{L}_{reg}}{\partial \widehat{W}^{(k)}}, \tag{1}$$

$$\frac{\partial \mathcal{L}}{\partial W^{(k)}} = \frac{\partial \mathcal{L}_{global}}{\partial W^{(k)}} + \alpha \frac{\partial \mathcal{L}_{local}}{\partial W^{(k)}} + \beta \frac{\partial \mathcal{L}_{reg}}{\partial W^{(k)}}, \quad k = 1, \ldots, K. \tag{2}$$

**Global proximity gradient.**  The global proximity loss is measured by reconstruction error:

$$\mathcal{L}_{global} = \sum_{i=1}^{m} \|\hat{\mathbf{z}}_i - \mathbf{z}_i\|_2^2.$$

Its gradient with respect to the decoder weight $\widehat{W}^{(k)}$ can be decomposed as

$$\frac{\partial \mathcal{L}_{global}}{\partial \widehat{W}^{(k)}} = \frac{\partial \mathcal{L}_{global}}{\partial \hat{Z}} \cdot \frac{\partial \hat{Z}}{\partial \widehat{W}^{(k)}}, \tag{3}$$

where

$$\frac{\partial \mathcal{L}_{global}}{\partial \hat{Z}} = 2(\hat{Z} - Z), \quad \hat{Z} = \sigma(\hat{Y}^{(K-1)} \widehat{W}^{(K)} + \hat{b}^{(K)}).$$

**Local proximity gradient.**  The local proximity loss is defined as

$$\mathcal{L}_{local} = \sum_{i,j=1}^{n} S'_{ij} \|\mathbf{y}_i - \mathbf{y}_j\|_2^2 = 2 \operatorname{tr}(Y^\top L Y),$$

where $L = D - S'$, $D$ is the diagonal degree matrix with $D_{ii} = \sum_j S'_{ij}$, and $Y$ is the embedding matrix. The gradient can be computed as

$$\frac{\partial \mathcal{L}_{local}}{\partial W^{(K)}} = \frac{\partial \mathcal{L}_{local}}{\partial Y} \cdot \frac{\partial Y}{\partial W^{(K)}}, \tag{4}$$

with

$$\frac{\partial \mathcal{L}_{local}}{\partial Y} = 2(L + L^\top)Y, \quad Y = \sigma(Y^{(K-1)} W^{(K)} + b^{(K)}).$$

**Training procedure.** DPGR combines the unsupervised component (reconstruction of neighborhoods to preserve global proximity) with the supervised component (local correlation preservation). These objectives are jointly optimized in a semi-supervised manner. To stabilize training and achieve effective initialization, the model is first pre-trained with a deep belief network Hinton (2009), followed by fine-tuning using stochastic gradient descent with backpropagation.

This optimization procedure ensures that DPGR captures both fine-grained local dependencies and broader global structures, yielding robust and discriminative embeddings for solver selection.

### A.1.2 CASCADE FOREST CONFIGURATION

For completeness, we provide additional details of the cascade forest implementation that were omitted in the main text. Each level of the cascade consists of two completely-random tree forests and two random forests, following the standard design of deep forest. Each forest contains 500 trees. In a completely-random tree forest, every split is made by randomly selecting one embedding dimension until pure leaves are reached. In contrast, in a random forest, the best split is chosen among $\sqrt{d}$ randomly sampled dimensions using the Gini index. These configurations remain fixed across all experiments.

## A.2 EXPERIMENTAL DETAILS

### A.2.1 DATASET

Table 3 summarizes the selected benchmark scenarios, including the number of algorithms and instances. As noted in the main text, we excluded the BNSL scenario due to excessive information loss during transformation, which makes it unsuitable for solver selection. For consistency across domains, CSP and MIP scenarios were converted into CNF: CSP instances were Booleanized and encoded into clauses following Tamura et al. Tamura et al. (2010), while MIP instances were transformed by encoding bounded integer variables into binary form and translating linear inequalities into pseudo-Boolean constraints, which were further reduced to CNF as in Sheini and Sakallah Sheini & Sakallah (2005).

Table 3: Overview of the problem scenarios. $|A|$ and $|I|$ denote the number of algorithms and instances, respectively.

| Scenarios | Alias | $|A|$ | $|I|$ |
|---|---|---|---|
| *SAT scenarios* | | | |
| SAT03-16 INDU | Sora | 10 | 2,000 |
| SAT12-ALL* | Svea | 31 | 1,614 |
| *MaxSAT scenarios* | | | |
| MAXSAT-PMS-2016 | Magnus | 19 | 100 |
| MAXSAT-WPMS-2016 | Monty | 18 | 100 |
| *CSP scenarios* | | | |
| CSP-Minizinc-Obj-2016 | Caren | 8 | 100 |
| CSP-Minizinc-Time-2016 | Camilla | 22 | 9,720 |
| *MIP scenarios* | | | |
| MIP-2016 | Mira | 5 | 218 |

For completeness, we briefly summarize the transformation principles. CSP instances can be encoded into SAT by Booleanizing domain variables and translating constraints into clauses, following the approach of Tamura et al. Tamura et al. (2010). MIP instances are transformed by representing bounded integer variables in binary form and converting linear inequalities into pseudo-Boolean constraints, which can be further encoded into CNF as in Sheini and Sakallah Sheini & Sakallah (2005). These transformations allow all selected scenarios to be consistently represented in SAT form.

### A.2.2 BASELINES

- **AS-ASL** Malone et al. (2017): A SAT solver selection method that uses Auto-sklearn to identify key features and train a stacking model for solver selection.

- **AS-RF** Malone et al. (2017): A SAT solver selection method that uses random forests for selecting the optimal solver based on problem instance features.

- **ASAP.v3** Gonard et al. (2017): A system that uses a sequential scheduler and algorithm selector to optimize solver selection, with a pre-scheduler identifying easier instances and selecting the best solver for more complex ones.

- **star-zilla** Xu et al. (2012): A solver portfolio that uses predictive modeling to select the best SAT solver from a set of candidates, demonstrating the efficacy of ensemble-based approaches.

- **sunny-as2-fk** Liu et al. (2021): A method that combines feature selection with k-nearest neighbor configuration to optimize SAT solver selection, improving performance by jointly tuning the neighborhood size and relevant features.

### A.2.3 EVALUATION METRICS

To evaluate the effectiveness of algorithm selection methods, we use the *gap* metric, which compares the performance of the selection system with the virtual best solver (VBS) and the single best solver (SBS). Formally, it is defined as

$$gap = \frac{m_{SBS} - m_s}{m_{SBS} - m_{VBS}}, \tag{5}$$

where $m_s$ denotes the performance of the selection system, $m_{SBS}$ is the performance of the SBS (the single solver that performs best on average across all instances), and $m_{VBS}$ is the performance of the VBS (the oracle that always selects the best solver for each instance).

The gap value ranges from $-\infty$ to 1:
- $gap = 1$ means the selection system performs as well as the VBS, i.e., it always selects the best solver for every instance.
- $0 < gap < 1$ indicates that the system improves over the SBS but has not yet reached the performance of the VBS.
- $gap = 0$ means the system only matches the SBS, i.e., no better than simply using a single solver across all instances.
- $gap < 0$ indicates that the system performs worse than the SBS, i.e., its selections on average lead to higher cost than always choosing the single best solver.

### A.3 ADDITIONAL EXPERIMENTAL RESULTS

### A.3.1 DETAILED ABLATION RESULTS

Figure 4 illustrates the precision comparison of F2S3-DS, F2S3-S, F2S3-D, and F2S3 across five random dataset samplings. Overall, the F2S3 model outperforms the other models in all 10 experiments, particularly demonstrating significantly higher precision rates than F2S3-DS, F2S3-S, and F2S3-D in most scenarios, indicating a clear advantage in task performance. Although the F2S3-D model performs closely, it still falls slightly short of F2S3, suggesting that the additional enhancements in the F2S3 model are effective in improving precision. The overall performance of F2S3-S and F2S3-DS is relatively lower, especially the F2S3-DS model, which shows markedly lower precision rates than the others in most scenarios. The differences in performance across various scenarios are also noteworthy. In certain scenarios, such as Magnus and Svea, the precision rates of F2S3 and F2S3-D are particularly prominent, indicating that these two models can better capture the features of these scenarios, exhibiting higher adaptability. In contrast, in scenarios like Camilla and Sora, the precision rates of all models are closer, yet F2S3 still slightly outperforms, demonstrating its consistent advantage across multiple types of scenarios. Furthermore, examining the fluctuations in the results of the five experiments, F2S3 shows a more stable performance across different random datasets with a smaller range of precision fluctuations, indicating strong robustness in multiple experiments. In comparison, F2S3-DS and F2S3 exhibit greater precision fluctuations across different experiments, especially showing instability in scenarios with lower precision rates. This further emphasizes the superiority and consistency of F2S3 under conditions of random datasets.

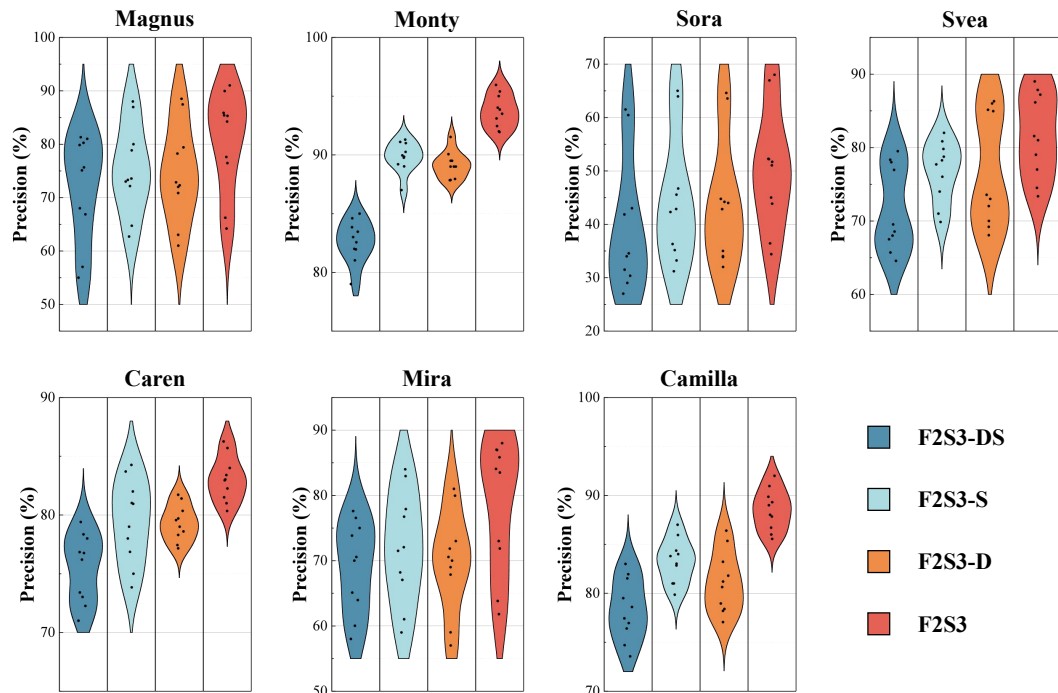

Figure 4: Results of ablation experiments for 10 runs.

### A.3.2 PARAMETER SENSITIVITY ANALYSIS

Parameter sensitivity analysis is a crucial step in understanding the relationship between model performance and key parameters. In this study, we focused on analyzing the impact of embedding dimension, hyperparameters $\alpha$ and $\beta$ on the performance of the network embedding model.

The investigation into embedding vector dimensionality reveals a notable influence on model performance. As depicted in Figure 5(a), performance initially rises and then falls with increasing dimensionality. This indicates that while an optimal dimension enhances information encoding, excessive dimensions introduce noise and degrade performance. Although our method shows low sensitivity to dimensionality, selecting an appropriate dimension is still essential.

The hyperparameter $\alpha$, which balances direct and indirect similarities, exhibits scenario-dependent optimal values as shown in Figure 5(b). Generally, $\alpha ¿ 0.1$ yields superior performance, emphasizing the significance of both direct and indirect similarities. Notably, higher $\alpha$ values enhance performance in Monty, Magnus, Sora, and Svea scenarios, while Camilla and Caren scenarios benefit from $\alpha$ values between 0.05 and 0.1. In the Mira scenario, $\alpha$ monotonically increases within [0, 0.2], further highlighting the importance of these similarities.

The hyperparameter $\beta$, controlling the reconstruction weight of non-zero elements in the training graph, shows an initial increase and subsequent decrease in model performance with increasing $\beta$ across seven scenarios (Figure 5(c)). Optimal performance is typically achieved when $\beta \in [5,7]$, with Monty and Mira scenarios peaking at $\beta = 8$. This indicates that moderate $\beta$ values improve non-zero element reconstruction, while excessive values degrade performance. The F2S3 model's enhanced performance with higher $\beta$ is attributed to its balanced reconstruction of non-zero and zero elements and optimization of the CRFG.

In summary, the parameter sensitivity analysis provides valuable insights into how to adjust model parameters to achieve optimal performance. In practical applications, these parameters should be selected and adjusted reasonably based on specific scenarios and requirements.

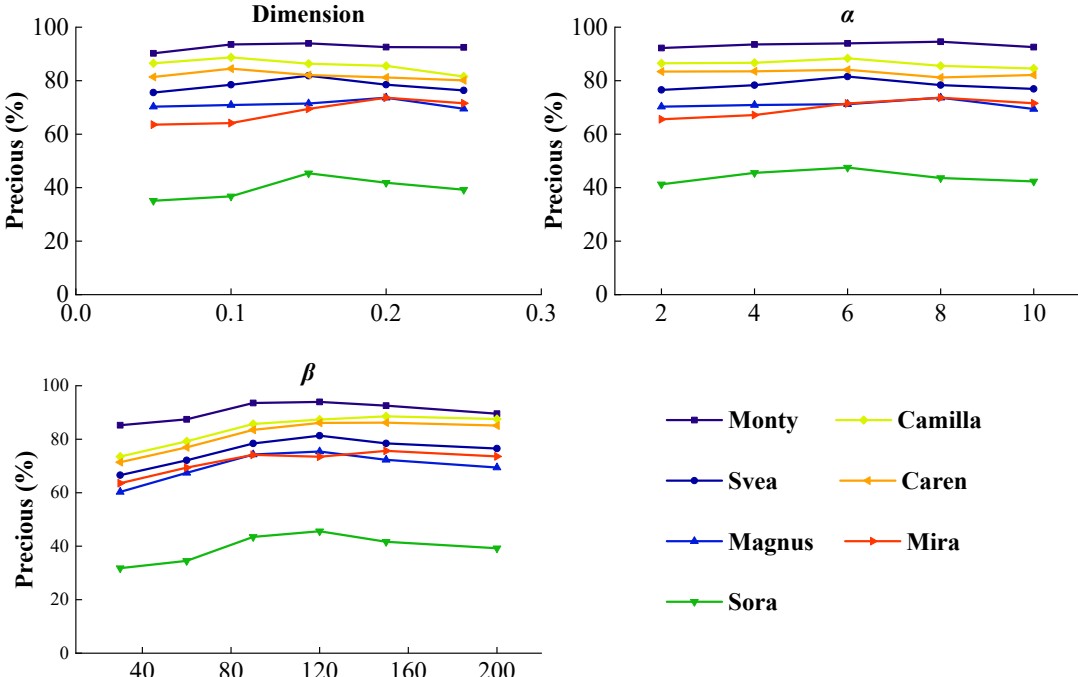

Figure 5: Parameter sensitivity comparison results. (a) Sensitivity comparison results for embedding dimension. (b) Sensitivity comparison results for parameter $\alpha$. (c) Sensitivity comparison results for parameter $\beta$.

