# OpenReview forum: "Feature-Free Approach for SAT Solver Selection"
_ICLR.cc/2026/Conference — ICLR 2026 Conference Withdrawn Submission_

### Official Review · Reviewer_hPjw · 2025-10-14

**Soundness:** 2
**Presentation:** 3
**Contribution:** 3
**Rating:** 4
**Confidence:** 4

**Summary:**

This work proposes a novel graph-representation of SAT problems by both a pre-processing step followed by a GNN-based autoencoder framework. The work also proposes an improvement on cascaded forests in order to use the learned representation for learning SAT selection. Empirical results are provided demonstrating that the full method's performance is generally better than baselines. Ablations are provided to demonstrate the contribution of each module of the method.

**Strengths:**

The paper is laid out in a mostly clear way
Figures are clear, results are easy to read
The graph representation, if I understood it correctly as a sum of weighted variable-clause and variable/clause incidence graphs, is interesting.

**Weaknesses:**

1. SAT competition data, which is generally the gold-standard data in terms of variety and difficulty, is not tested on
2. GRASS, which is included in the related works section as the most recent graph-based solver selection method, is not included as a baseline
3. Looking at GRASS, it reports raw time improvement and classification accuracy. I think these are both more interesting metrics than GAP, since the order of magnitude of the time-saved is informative of incremental value of the model relative to its baselines.
4. There where some unclarities in the writing, which I list in the Questions section below.
5. The graph neural networks section of the related works is incomplete - the SAT generation field is widely concerned with learning graph representations. In fact, your autoencoder is fundamentally a generative model; this further motivates inclusion of the generative literature. See [1], [2], [3], [4], [5]
6. Some text formatting issues, most notably that the in-text citations are not done parenthetically. Authors should use \citep when citing an article of the form "NeuroSAT (Selsam et al., 2019) first demonstrated the potential of message-passing", and \citet when citing an article in the form "Selsam et al. (2019) propose NeuroSAT, which first demonstrated the potential of message-passing"




[1] Yang Li, Xinyan Chen, Wenxuan Guo, Xijun Li, Junhua Huang, Hui-Ling Zhen, Mingxuan Yuan, and
Junchi Yan. 2023. HardSATGEN: Understanding the Difficulty of Hard SAT Formula Generation
and A Strong Structure-Hardness-Aware Baseline. In Proc. of ACM SIGKDD Conf. Knowledge
Discovery and Data Mining.

[2] Weihuang Wen and Tianshu Yu. 2023. W2SAT: Learning to generate SAT instances from Weighted
Literal Incidence Graphs. arXiv:2302.00272 [cs.LG]

[3] Jiaxuan You, Haoze Wu, Clark Barrett, Raghuram Ramanujan, and Jure Leskovec. 2019. G2SAT:
Learning to Generate SAT Formulas. In Proc. Adv. Neural Inf. Process. Syst.

[4] Cotnareanu, Joseph, et al. "HardCore Generation: Generating Hard UNSAT Problems for Data Augmentation." Advances in Neural Information Processing Systems 37 (2024): 62409-62431.

[5] Chen, Xinyan, et al. "Mixsatgen: Learning graph mixing for sat instance generation." The Twelfth International Conference on Learning Representations. 2024.

**Questions:**

1. A major obstacle which deep-learned SAT models face is heterogeneity: as datasets become constructed from more distinct families of problems, the learning breaks down. NeuroSAT/NeuroCore (https://arxiv.org/abs/1903.04671), GRASS, and [4] (above) discuss this. Your claim that features are expensive to compute is tenuous, since Grass reported ~1s feature compute time per problem. My question is will your featureless representation generalize better to settings with heterogeneous constructions such as the SAT-Competition than existing feature-based methods? If yes, this would be a much stronger motivation for your representation learning than feature cost. A simple benchmarking of your method against SAT-Comp would answer this
2. l. 223: "bipartite adjacency", is the graph really bi-partite since you're connecting variable nodes to each-other based on their incidence and similarity? Same with clause nodes.
3. l. 223: In fact, in your definition of S', clauses and variables are never connected. Are you maybe doing S' = S + Direct + Indirect?
4. l. 223: "Optimized" Can you describe the optimization process to obtain CRFG? You don't describe an objective function or optimization procedure for this section, only for the DPGR section. I can't tell which parameters are being optimized; is it N(\cdot)?
5. l. 262: Global proximity preservation: does this objective really ensure global structural learning? By comparing the corresponding input/output neighborhoods, is this not a local measure?
6. With enough layers, MPNNs can achieve the information propagation which you claim is restricted by sparsity. Normally, graph-smoothing is an obstacle which limits the number of MP layers one might add to an MPNN-based model. With that said, the presented setting is a graph-level prediction one, in which node-level smoothing is not necessarily an issue. So, the claim at l. 193 that local propagation limits long-hop information propagation seems tenuous. Can you provide some additional argumentation or experimental results to support this claim?

If questions 2, 3, and 4 are the results of poor word-choice or mistakes in notation which can be quickly fixed, then please clarify and describe how you will correct the relevant text for the camera-ready and I will raise my score (assuming you also address the described weaknesses above). If not, without some further discussion to convince me of why I might be wrong in my understanding, I can't recommend that this paper be accepted.

---

### Official Review · Reviewer_Zirb · 2025-10-26

**Soundness:** 2
**Presentation:** 2
**Contribution:** 2
**Rating:** 2
**Confidence:** 3

**Summary:**

The paper introduces F2S3, a feature-free SAT solver selection framework that leverages the power of Graph Neural Networks (GNNs) to automatically learn which solver to apply for a given problem instance. The authors specifically introduce Correlation Refinement Factor Graphs (CRFGs), Dual-Proximity Graph Representations (DPGRs), and Sensitivity-Association Cascade Forest (SACF) to effectively capture the structural properties of SAT instances. By transforming various combinatorial problems into SAT problems, F2S3 can be applied to a wide range of domains. The proposed method is evaluated on multiple benchmarks that are deployed in the OASC 2017 competition dataset, and demonstrates superior performance compared to existing algorithm selection methods.

**Strengths:**

- The experimental results show that their proposed method outperforms other algorithm selection methods on multiple benchmarks.
- The idea of transforming various combinatorial problems into SAT problems to allow the use of GNNs for algorithm selection is interesting and novel.
- The ablations support that DPGR and SACF contribute to the overall performance.

**Weaknesses:**

- In L473, they mention that CRFG, DPGR and SACF complement each other, but in their ablation studies, they only look at DPGR and SACF, and do not provide any ablation studies for CRFGs to support this claim.
- The OASC 2017 results show that ASAP.v2 was the overall winning method (better than v3), but the authors do not include this in the experimental results without any explanation.
- There are a lot of techniques combined in this work, making it hard to assess the contribution of each individual component.
- The overhead for selecting algorithms is not discussed in the experiments at all.
- The results reported seem mostly identical to the OASC 2017 results (aside from Camilla), but there is no discussion on how the experimental results are obtained, and the repository given do not provide any code for reproducing the baseline results.
- Following up on the previous point, my main concern is that the official OASC 2017 rules that the authors seemingly reference/reproduce imply that the competitors are not provided access to the problem instance directly. If so, this makes it completely unfair for other methods that only get access to features provided by the organizers, in comparison to F2S3 which, by design, requires full access to the problem instance. I am happy to be corrected on this point if I misunderstood the rules or the framework, but if this is indeed the case, this is potentially a major flaw in the experimental settings. For reference, [1] states that "the participants had access to performance and feature data on training instances (2/3), and only the instance features for the test instances (1/3)."

[1] Lindauer, M., van Rijn, J. N., & Kotthoff, L. (2017). Open Algorithm Selection Challenge 2017: Setup and Scenarios. *Proceedings of the Open Algorithm Selection Challenge 2017, Brussels, Belgium, September 11-12, 2017*. Proceedings of Machine Learning Research, 79, 1–7.

Formatting concerns:
- Citations are not formatted correctly; they are missing parentheses. For example SATzilla Xu et al. (2008) at L123 should be SATzilla (Xu et al., 2008).

**Questions:**

- How does the inference time of F2S3 compare with other algorithm selection methods? I imagine that transforming other domain problems into SAT problems may add significant overhead as well.
- In prior works, they refer to the metrics as *closed gap* scores, but the paper says gap values. Is there a difference between these two metrics? Also, is there a reason why you did not choose the different version of the gap score as shown in [1]?
- I think the feature-free claim is not very accurate; while the method does not rely on hand-crafted features, it still relies on features (embeddings) extracted by the GNN. Could the authors clarify this point?
- I imagine that there are problems that are very large. Is F2S3 able to handle such problems, or does it have limitations in terms of problem size?
- The scores given for Camilla seem to be completely different from the results of the OASC 2017 report. Could the authors clarify what the differences are here?

[1] Lindauer, M., van Rijn, J. N., & Kotthoff, L. (2019). The algorithm selection competitions 2015 and 2017. *Artificial Intelligence, 272*, 86–100.

---

### Official Review · Reviewer_8FnU · 2025-10-27

**Soundness:** 2
**Presentation:** 2
**Contribution:** 2
**Rating:** 2
**Confidence:** 3

**Summary:**

The paper is essentially trying to make a better portfolio-based SAT approach. The idea is to match SAT instances to their optimal solvers better at test time.

The *claim* is that their approach gets rid of a lot of manual engineering / feature picking that currently pervades the field. The results are essentially not very conclusive, in my opinion.

**Strengths:**

I found the paper clear and easy to follow. The literature is mostly presented accurately and I believe someone without familiarity of SAT solvers can follow it as well. The figures / tables / presentation etc. is clear.

**Weaknesses:**

I am not at all convinced of the empirical strength of the paper. In the field of SAT solving, contests are held every year, and it is usual to eval against other methods on this benchmark. Neither is this used as a dataset, nor are the recent methods that come out in this field and in fact are part of the lit review, used as peers ! It is not possible to evaluate the argument empirically.

Is there theoretical merit ? Here too I believe the argument is weak. The paper seeks to get rid of manual feature making, and so on. But its approach is significantly contrived, and indeed I do not see how some of the choices are just not manual feature making by another name ("manual function making", almost). So, currently, the paper has immense weaknesses.

**Questions:**

Please provide a comparison of the method on a recent SAT contest against at least a few SOTA/near methods from *recent* years. Otherwise it's not easy to be sure if the method works or not.

---

### Official Review · Reviewer_Hsfu · 2025-10-29

**Soundness:** 3
**Presentation:** 3
**Contribution:** 3
**Rating:** 6
**Confidence:** 3

**Summary:**

This paper proposes a graph-based method for SAT solver selection, leveraging learned representations of SAT instances to predict which solver performs best on a given instance. The core idea is to encode SAT formulas as bipartite graphs (variables-clauses) and apply a graph neural network (GNN) to extract instance-level embeddings. These embeddings are then used to compute similarity between instances, enabling nearest-neighbor-based solver recommendation. The authors report promising empirical results on standard benchmarks, showing improved solver selection accuracy over existing feature-based approaches.

**Strengths:**

1. The work presents a reasonably original application of GNNs to the classical problem of SAT solver selection.

2. The technical approach is sound and well-motivated. The use of a bipartite graph representation and message-passing GNNs is appropriate for capturing structural properties of SAT instances.

3. The paper is generally well-written and easy to follow.

**Weaknesses:**

1. The empirical results are limited to a major experiment and an ablation study. This paper can benefit from more benchmark evaluations with different scales and more analyses, like visualization of features and the selection ratio of different kinds of solvers.

2. The authors claim that previous works mainly adopt hand-crafted features for solver selection. If so, naive graph learning baselines (e.g., using simple GNNs like GCN or GAT without their proposed similarity mechanism, or even unsupervised graph embeddings) should be included to better demonstrate the lift from the proposed designs.

3. The paper omits experimental details like the size (number of variables/clauses) of the SAT instances used and the solvers constituting the candidate portfolio.

**Questions:**

1. The method uses a weighted combination of two similarity metrics (e.g., embedding similarity and runtime similarity). How is the trade-off parameter chosen? Is it tuned per dataset, or fixed? Does performance vary significantly with this parameter across different problem domains?

2. Could the authors clarify:
- The typical size (e.g., median number of variables/clauses) of instances in the evaluation?
- The exact set of solvers in the selection portfolio (e.g., MiniSat, Glucose, MapleSAT, etc.)?

3. Would the authors consider adding a baseline that uses the same GNN to directly predict solver performance (e.g., via regression or ranking loss), rather than relying on nearest-neighbor similarity? This would help isolate whether gains come from the representation or the selection mechanism.

---

### Note · Authors · 2025-11-17

I have read and agree with the venue's withdrawal policy on behalf of myself and my co-authors.